# Histomorphology and Chemical Constituents of Interdigital Gland of Vembur Sheep, *Ovis aries*

**DOI:** 10.3390/vetsci9110647

**Published:** 2022-11-21

**Authors:** Thangavel Rajagopal, Selvam Mahalakshmi, Thirukonda Ravindhran Gayathri, Naganathan Muruganantham, Marimuthu Muthukatturaja, Durairaj Rajesh, Kamatchi Rameshkumar, Ponnirul Ponmanickam, Mohammad Abdulkader Akbarsha, Govindaraju Archunan

**Affiliations:** 1Department of Zoology, Thiagarajar College (Autonomous), Madurai 625009, Tamil Nadu, India; 2Research Institute in Semiochemistry and Applied Ethology (IRSEA), Quartier Salignan, 84400 Apt, France; 3Department of Zoology, Rajah Serfoji Government College (Autonomous), Thanjavur 613006, Tamil Nadu, India; 4Department of Zoology, Ayya Nadar Janaki Ammal College (Autonomous), Sivakasi 626124, Tamil Nadu, India; 5Department of Biotechnology & Research Coordinator, National College (Autonomous), Tiruchirappalli 620001, Tamil Nadu, India; 6Pheromone Technology Lab, Department of Animal Science, School of Life Sciences, Bharathidasan University, Tiruchirappalli 620024, Tamil Nadu, India; 7Research Dean, Marudupandiyar College, Thanjavur 613403, Tamil Nadu, India

**Keywords:** Vembur sheep, interdigital gland, histomorphology, apocrine secretory lobules, sebaceous secretory lobules, volatile compounds

## Abstract

**Simple Summary:**

This study aimed at finding the male-female differences in the anatomy, morphology, and volatile compounds of the interdigital glands of a South Indian breed of Vembur sheep. These are skin glands located between the digits. The glands resemble a tobacco pipe and consist of a body, flexure, and an excretory duct with an external orifice located between the digits. There are three distinct layers of tissues, epidermis, dermis, and fibrous capsule. Apocrine and sebaceous glandular lobules are present in the dermis in both sexes, but in the males the lobules are in higher densities and diameters than females. As many as 23 major compounds were identified in the interdigital gland postings of male and female sheep. Four of them were present only in the male glandular post, whereas five others were present only in the female glandular post. Both male and female sheep had appreciable levels of compounds that are antibacterial substances in pronghorn antelopes. It is concluded that the interdigital gland of Vembur sheep releases a variety of chemical compounds from the two secretory lobules, which may serve as chemical communication systems and also protect the sheep from foot-borne diseases.

**Abstract:**

The interdigital gland is a specialized skin gland located between the digits of Artiodactyla (i.e., even-toed ungulates). Its secretion participates in semiochemical communication, and protects from ultraviolet radiation as well as fungal and bacterial infections of the feet. The present study aimed at finding if there are male-female differences in the anatomy, morphology, and volatile compounds of the interdigital gland of the South Indian breed of Vembur sheep. A total of 24 sheep (12 each of male and female) were spotted at the slaughterhouse and the interdigital gland was removed for examination. The anatomical examination revealed it to resemble a tobacco pipe and to consist of a body, flexure, and excretory duct with an external orifice located at the cleft of the digits. Morphometrically, the interdigital glands differed between males and females. The gland possesses a distinct fibrous capsule, epidermis, and dermis. The fibrous capsule contains several parallel bundles of collagen fibers, nerve fibers, and blood vessels, etc. The epidermis consists of keratinized squamous epithelium formed of stratum basale, stratum granulosum and stratum spinosum. The dermis consists of hair follicles, nerve plexuses, arrector pili muscles, and apocrine and sebaceous glandular lobules. The latter, lined by a simple cuboidal epithelium, are arranged in clusters of acini in the upper portion of the dermis. The apocrine secretory lobules, made up of parenchymal cells, are found in the lower portion of the dermis. The density and diameter of the apocrine and sebaceous secretory lobules were significantly higher in the males than females. Scanning electron microscopic (SEM) analysis confirmed the apocrine and sebaceous secretory components. Twenty-three major compounds were identified in the interdigital gland postings of male and female sheep, among which butanoic acid, 2-methylpropanoic acid, 1-heptanol and octadecanoic acid were present only in the male glandular post, whereas octane, 7-hexyl-tridecane, tetradecane, heptadecane and decanoic acid were present only in the female glandular post. Tetradecanol, tetradecanoic acid and hexadecanol peaks, reportedly antibacterial compounds in pronghorn antelopes, were highly prominent in both male and female sheep. Thus, the interdigital gland of Vembur sheep has two major secretory lobules, namely, sebaceous and apocrine, larger in males than females, which secrete a variety chemical compounds that may serve as chemical communication systems and protect the sheep from foot-borne diseases.

## 1. Introduction

The Indian Council for Agricultural Research (ICAR) data indicates that there are 40 different breeds of sheep in India. Tamil Nadu, one of the geographic and political states, has 7.99 million sheep, constituting 11.17 percent of the country’s total sheep population, and ranks fourth in India. There are eight recognized sheep breeds in Tamil Nadu, three wool types, viz., Coimbatore, Nilgiri, and Tiruchi Black, and five meat types viz., Kilakarsal, Madras Red, Mecheri, Ramnad White, and Vembur. They are scattered throughout the southern agroclimatic zone of Tamil Nadu [1,2]. High levels of endemism and regional variations in agroclimatic conditions (soil condition, climate, rainfall, humidity, and grazing facilities) have contributed to the formation of numerous breeds and strains of sheep in India, which have a high degree of environmental adaptation. Most of these breeds are given their names based of where they live; however, some are designated as distinctive traits. The indigenous sheep are an essential component of the agrarian economy, particularly in areas where crops and dairy farming are not economically viable, and they play a major role in the livelihoods of a large proportion of small and marginal farmers, as well as landless laborers. However, due to interbreeding, exotic breed introductions, and changes in farming systems, the purebred population has declined and genetic merit has been diluted [1].

The Vembur sheep are an important livestock breed in Tamil Nadu, as they provide meat, fiber, and skin to the poor, and are often referred to as the “poor man’s cow”. Since the beginning of the Tamil culture, these sheep have been used for agricultural, economic, cultural, and even religious purposes. This sheep is found in three districts of southern Tamil Nadu, namely Thoothukudi, Virudhunagar, and Tirunelveli. It is the most versatile and widely dispersed livestock species in India, living everywhere from high altitudes to deserts and humid regions [1,2,3]. Since sheep yield a variety of products of commercial value, and have a relatively short gestation period, they are an ideal domestic species for biological studies and applications [4].

The even-toed ungulates (Artiodactyla) possess many specialized skin glands, the secretions of which participate in semiochemical communication [5]. The following scent glands are present in even-toed ungulates: sudoriferous glands between the antlers and eyes on the forehead; preorbital glands extending from each eye’s medial canthus; nasal glands found between the nostrils; preputial glands found in the foreskin of the penis; interdigital glands found between the toes; the metatarsal glands positioned outside the hind legs; the tarsal glands located within the hind legs; and the inguinal glands placed in the lower belly or groin [5]. The interdigital glands are very prominent scent glands that produce pheromones in sheep, serow, and goats, which serve a variety of vital roles such as demarcation of territorial areas, sexual communication, and the expression of social behavior [6,7,8]. This gland is a specialized skin gland located between the digits and contains sebaceous and apocrine secretory lobules [9,10,11,12,13]. According to Bacha and Bacha [14], the interdigital gland contains numerous sebaceous and apocrine secretory glands that discharge waxy substances through an orifice above the foot and serve as scent marking. In addition, the interdigital gland is responsible for maintaining the elasticity of the skin in the region where it is located; its secretion possesses anti-fungal and anti-bacterial properties that protect the foot from poor hygienic conditions, high humidity during spring, and mechanical injury [6,14,15,16]. Jaber and Mazeg [17] found that the glandular oily secretion can resolve infection and inflammation, without the need for antibiotics and anti-inflammatory medications.

The anatomy and histomorphology of the interdigital gland have been studied in several species of mammals, including sheep [7,8,16,18,19,20], goat [21,22,23], serow [11], and roebuck [24] etc. The chemical constituents of glandular secretion have been only sparingly analyzed in mammals including reindeer [25], black-tailed deer [26], sable antelope [27], bontebok [28], whitetail deer [29], and pronghorn [30], etc. However, the anatomy, histomorphology and the chemical constituents of interdigital glands of south Indian breeds of Vembur sheep have not been studied so far [13]. Therefore, the present investigation was undertaken to examine the differences between males and females in respect of the morphometry and histomorphology of the interdigital gland and the chemical composition of its secretion in Vembur sheep, *Ovis aries*.

## 2. Materials and Methods

### 2.1. Study Animals

The Vembur sheep (Figure 1) are native to the Vembur village of the Pudur Taluk, in the Thoothukudi district of Tamil Nadu (8.62° N and 77.97° E). They are tall animals with irregular black and fawn patches all over their white bodies. The tail is thin and short, and the ears are medium in size and drooping. The ewes have polls, whereas the males have horns. The body is covered with short hair, which is not shorn. The lambing percentage of farmers’ flocks is 80 and the litter size is single. Breeding is generally performed on the basis of selection from within the flock. There are two breeding seasons in the Vembur breeding tract: most of the animals are in heat from March to May, and a few are so from July to September.

### 2.2. Source of the Interdigital Gland

The samples of interdigital gland used in the study were collected from Vembur sheep that were slaughtered in a private slaughterhouse in Madurai, Tamilnadu, India. The glands of the forefeet and hind feet of 12 each of male and female Vembur sheep (1–2 years old) were examined in terms of morphometry, histomorphology, and volatile identification. The feet of six each of male and female sheep were used for gross dissection, morphometric measurements, and histological examination, while the feet of the remaining six each were used for characterization of volatiles using GC-MS.

### 2.3. Morphometric Examinations

The morphometric parameters (weight, length, and width) of the interdigital gland were measured using a digital caliper (Mitutuyo Corporation, Kawasaki, Kanagawa, Japan) and the images of the gland were recorded in a digital camera (Canon EOS 650D; Tokyo, Japan).

### 2.4. Histological Study

Tissues fixed in 10% formalin were washed in running tap water and subjected to dehydration through a graded series of alcohol, followed by clearing in xylene, and then embedded in paraffin wax. Transverse and longitudinal sections at 3–5 μm thickness were obtained using a rotary microtome (Leica, Wetzlar, Germany). The sections, thus obtained, were stained in Harris’ hematoxylin and eosin, dehydrated using alcohol, cleared in xylene, and mounted in DPX adhesive resin. Sections were examined in a light microscope (Olympus BX51, Tokyo, Japan).

### 2.5. SEM Analysis

The interdigital glands of both male and female sheep were evaluated using scanning electron microscopy (SEM). Samples of the interdigital gland/sinus, sliced transversely or longitudinally, were pre-fixed in 3% glutaraldehyde-PBS (pH 7.2) at 4 °C for 2 h. Following several washes in PBS, the samples were kept in 1% osmium tetroxide-PBS for 2 h. The samples were then washed overnight and dehydrated through serial dilutions of ethanol. The samples were mounted onto stubs, sputter-coated with gold by Polaron SC-500 (Microtech, East Sussex, Wadhurst, England) and finally examined in a JSM 5600 JEOL scanning electron microscope (Jeol Co., Tokyo, Japan) and photographed.

### 2.6. Sample Preparation for GC-MS Analysis

Samples of interdigital glandular secretion were collected from male and female Vembur sheep for chemical analysis. Initially, individual male and female samples were analyzed separately by adopting GC-MS to find the chemical profiles, and then samples from males and females were pooled separately for GC-MS analysis.

A portion (1 g) of the post material (i.e., interdigital gland secretion) was homogenized with PBS (pH 7.2) separately for males and females. The homogenate was clarified by centrifugation at 2500× *g* for 5 min and the supernatant was removed and mixed with 1 mL DCM (1:1 ratio) and filtered through a silica gel column (60–120 meshes) for 30 min at 37 °C. The filtered extract was cooled with liquid nitrogen to condense it to 1/5th of its original volume.

### 2.7. Identification of Volatile Compounds by GC-MS

The extract (2 µL) was injected into the GC-MS apparatus on a 30 m glass capillary column with a film thickness of 0.25 m (30 m × 0.2 mm i.d. coated with UCON HB 2000) at the following temperatures: Initial oven temperature 40 degrees Celsius for 4 min, then escalated to 250 degrees Celsius at 15 degrees Celsius for 15 min, and finally kept at 250 degrees Celsius for 10 min. The GC unit had a FID detector that was connected to an integrator. Each component’s relative amount was expressed as the percentage of the total ion current. The GC-MS was controlled by a computer at 70 eV, and ammonia was used as the reagent gas at 95 eV. Probability-based matching with the computer library built within the NICT 12 system identified the unknown compounds.

### 2.8. Statistical Analysis

The weight, length, and width of interdigital glands are presented as means ± standard errors. The paired sample *t*-test was performed to analyze the variations in male and female interdigital glands. Statistical differences were considered significant at *p* < 0.05. In addition, the density and diameter of the sebaceous and apocrine glands were measured by using a calibrated ocular micrometer at ×450 according to the modified method described by Haffner [31].

## 3. Results

### 3.1. Anatomical Features

In both fore- and hind limbs, the interdigital gland is found in the space between the digits at the proximal and medial phalanges, and its opening is located just above the interdigital cleft (Figure 2A,B). The gland is dull-white in color and resembles a teapot or a tobacco pipe because it has a bend connecting the two portions and an orifice at the proximal end of its wide body and the long narrow neck (i.e., pouch and narrow neck). As revealed in the dissections, the pouch-like portion is secretory and the narrow canal is the excretory duct. The duct comprises four distinct parts viz., the orifice, excretory duct, flexure and corpus (Figure 2C,D). The lumen (i.e., the pouch) of the gland is filled with a dense secretory material. There are numerous wooly fibers embedded in the luminal content and there is a dense oily material at the opening of the gland which emanates a foul and rancid odor.

### 3.2. Morphometric Data

The morphometric measurements of the interdigital gland are shown in Figure 3. The weight, length, and width of the gland were found to greatly vary between male (weight: 8.5–6.5 g; length: 2.8–3.5 cm; width: 1.5–1.7 cm) and female sheep (weight: 3.5–5 g; length: 2.0–2.5 cm; width: 0.8–1.5 cm). The paired *t*-test showed that the mean weight, length, and width of the interdigital gland are significantly (weight: *t* = 10.5 mg, df = 5; length: *t* = 4.64 cm, df = 5; width: *t* = 4.35 cm, df = 5, *p* < 0.05) higher in males (weight: 7.5 ± 0.39 mg; length: 3.05 ± 0.14 cm; and width: 1.61 ± 0.03 cm) than females (weight: 4.25 ± 0.23 mg; length: 2.36 ± 0.07 cm; and width: 1.15 ± 0.09 cm). Further, no variation in the measurements of the glands between the right and left feet was noticed, and one in the fore-feet was larger than that in the hind-feet.

### 3.3. Histological Features

Both the pouch (wide body) and the neck (narrow duct) of the interdigital gland are covered by three distinct layers: fibrous capsule, epidermis, and dermis (Figure 4). The fibrous capsule is the outermost part of the interdigital sinus. It is composed of dense connective tissue, containing several parallel bundles of collagen fibers, adipose tissue, nerve fibers, blood vessels, etc. The epidermis consists of a stratified squamous epithelium and the mucosal folds that project into the lumen. In addition, the stratum granulosum is well-developed, whereas the stratum papillare is prominent and contains loose connective tissue and elastic fibers. Melanin granules are observed in the stratum basale (Figure 5). The dermis layer is composed of hair follicles, arrector pili muscles and apocrine and sebaceous secretory lobules of different sizes. The upper portion of the dermis is occupied by sebaceous secretory lobules, hair follicles, and arrector pili muscles, whereas the lower portion is occupied by clusters of apocrine secretory glands (Figure 6). The sebaceous secretory glands are always associated with hair follicles (Figure 7). The glandular secretions (both apocrine and sebaceous) empty directly onto the narrow excretory duct that opens into the external orifice (Figure 8).

The secretory cells of apocrine lobules are composed of a single layer of cuboidal cells resting on a basement membrane and surrounded externally by myoepithelial cells and a thin layer of connective tissue fibers (Figure 9). The secretory cells of the apocrine secretory lobules are lined by simple to stratified cuboidal cells. The lumen of the apocrine secretory lobule has interdigitating sinuses of different shapes and sizes but, generally, simple coiled tubular type.

The interdigital glandular tissue of male and female Vembur sheep contains normal/ordinary holocrine sebaceous secretory lobules. The secretory acini (i.e., serous, and mucous acinar cells) of the holocrine sebaceous glands are tear-shaped, with a clearly visible cell membrane and nucleus. The sebaceous glands empty directly on the skin surface and most of the secretory lobules open into the hair follicles. The combined sebaceous alveoli form a kind of excretory duct through which the secretion is discharged into the lumen of the pouch (Figure 10). The cells lie closer to the center of the alveoli and become progressively large. The cytoplasm is distended with fat droplets. The cells show a gradation of secretory activity from the deposition of single fat globules to the disintegration of the cells. The nuclei shrink in the process and finally disappear altogether. Cells lost in secretion are replaced by the peripheral cells. When the fat of the cells is extracted using a solvent, the cytoplasm becomes honeycomb in appearance (Figure 11).

### 3.4. Density and Diameter of Apocrine and Sebaceous Lobules

Microscopic examination revealed that male and female Vembur sheep’s interdigital glands differ in the density and diameter of apocrine and sebaceous secretory glands. As revealed in the paired sample *t*-test, the apocrine and sebaceous gland densities are significantly (AG: *t* = 2.61, df = 5; SG: *t* = 3.33, df = 5; *p* < 0.05) higher in male glands (AG: 47.5 ± 2.83 units/mm^2^; SG: 28.83 ± 1.42 units/mm^2^) than in female glands (AG: 34.86 ± 2.81 units/mm^2^; SG: 21.66 ± 1.41 units/mm^2^). Further, the diameter of apocrine and sebaceous secretory lobules varied significantly (AG: *t* = 4.75, df = 5; SG: *t* = 5.82, df = 5, *p* < 0.05) in the male gland (AG: 6.22 ± 0.31 µm; SG: 8.01 ± 0.40 µm) compared to female gland (AG: 4.65 ± 0.26 µm; SG: 5.81 ± 0.20 µm).

### 3.5. SEM Analysis of Interdigital Gland

Scanning electron microscopy (SEM) was adopted to evaluate the interdigital glands (Figure 12). The gland shows two glandular components: one displaying apocrine secretory lobules and the other with sebaceous secretory units (Figure 12A). The apocrine secretory lobules are quite tall and often exhibit apocrine blebs, designating the pinching off of secretion to the lumen (Figure 12B). The sebaceous secretory lobules are positioned clustering the inner surface and appear like a group of bubbles (honeycomb) and clusters of secretory acinar cells (Figure 12C). The apocrine lobules usually appear as a group of tortuous tubules, and their lumen is often occupied by secreted content and the secretory vesicles (Figure 12D).

### 3.6. GC-MS Profiles of Interdigital Glandular Post

The GC-MS profiles shown in Table 1 and Figure 13 are the compounds present in the interdigital glandular secretion of male and female Vembur sheep. The secretion showed 18–19 detectable peaks in males and females put together. However, the male and female sheep glandular secretions were found to reveal 23 volatile compounds, with 18 compounds found in the secretion of male sheep and 19 in female sheep. The different chemical constituents identified in the glandular posts were alkane, alkene, carboxylic acid, alcohol, and amine. Among the different constituents, alkanes were the most predominantly present in the glandular post samples. Comparison of the compounds between male and female sheep revealed that certain compounds are specific to a particular sex. For instance, among the 23 volatiles, four compounds—butanoic acid, 2-methylpropanoic acid, 1-heptanol and octadecanoic acid- were specific to male gland secretion and absent in female gland secretion. As many as 19 volatiles were identified as major compounds of the female gland secretion of which five—octane, 7-hexyl-tridecane, tetradecane, heptadecane and decanoic acid- showed to be specific for female glandular post. Put together, 14 volatile compounds were present in both male and female glandular secretion, among which the peak height concentrations of tetradecanol, tetradecanoic acid and hexadecanol were greatly high compared to the other compounds.

## 4. Discussion

The present study focused on the anatomical, morphological, and histological features and identification of volatile compounds (i.e., putative pheromones) of the interdigital gland of the South Indian breed of Vembur sheep. The gland is present in all four limbs in Vembur sheep, and is located between the proximal and distal interphalangeal joints of the two main digits. The location of the interdigital gland in our study is consistent with earlier reports in respect of Iranian sheep [6], Tuj sheep [7], Lameness sheep [16], Kivircik sheep [32], and Awassi sheep [33], etc. The shape of the interdigital gland reportedly resembles a tobacco pipe or teapot and consists of a body and a neck [8,32]. Abbasi et al. [6] described it as a narrow channel with a large pouch. Demiraslan et al. [32] described the gland as formed of a body (corpus or glandular part) and a neck, connected to the exterior by the external orifice [6,18,33]. Anatomical examination in this study revealed that the interdigital gland of Vembur sheep has a pipe-shaped structure consisting of three parts: a narrow neck with an orifice at the proximal end; a wide body; and a bend connecting these two parts.

The interdigital gland of male and female sheep differs significantly in morphometric parameters (i.e., length, width, and weight), which indicates a sexual difference. Moreover, males have higher values for interdigital gland weight, length, and width than females. These findings are in agreement with those reported by Abbasi et al. [6] in Iranian sheep, Rajagopal and Archunan [34] in Indian Blackbuck, Awaad et al. [35] in Egyptian sheep, Yilmaz et al. [33] in Awassi sheep, Ponmanickam et al. [36] in the Bandicoot rat, Kara et al. [18] in Hasmer and Hasak sheep, and Rajagopal et al. [37] in the soft-furred field rat, etc. In another context, the dominant male (adult) blackbucks have larger preorbital glands and higher testosterone levels than subordinate males (sub-adults and adolescents) [34,38]. Despite the scent glands (i.e., male preputial and female clitoral glands) of rats not exhibiting sexual differences, gonadectomy resulted in a significant decrease in their size in both sexes [39]. A study conducted by Mckinney and Desjardins [40] revealed that male dominant mice had higher testosterone levels and larger preputial glands than male subordinate mice. Moawad [41] reported that the higher development of preorbital glands in male fallow deer than female deer would depend on the higher social responsibility (dominance hierarchy)/production of testosterone. Therefore, the larger size and greater weight of the male interdigital gland may reflect the manifestation of androgen support. Thus, it is worthwhile to comprehend that the size of the interdigital glands is influenced by sex hormones, and that they function to convey the reproductive status to the opposite sex.

Microscopic examination of the interdigital gland revealed that it has a narrow neck and a body; the body has a larger lumen than the neck and it is covered by a fibrous capsule. The lumen of the neck and body parts are both filled with dull-white secretions and numerous broken fibrous capsules, which are made up of three separate layers viz., the fibrous capsule, epidermis, and dermis. There are well-developed apocrine and holocrine sebaceous secretory lobes in the dermis, where the sebaceous gland is significantly larger than the apocrine gland. Similar observations have been made on the interdigital gland of various other species, including the madoqua [42], red duiker [9], Kivircik sheep [32], Awassi sheep [33], Hasak sheep [18], and Konya Merino sheep [19] etc. In contrast, the preputial and clitoral glands of male and female soft-furred field rats show sexual dimorphism, the male having sebaceous and apocrine secretory lobules, whereas the female having apocrine secretory lobules [37]. The infraorbital gland of male Japanese serow has an ordinary sebaceous gland, whereas female has a modified one [43]. These findings suggest that the histoarchitectural features of interdigital glands of Vembur sheep have well-developed sebaceous and apocrine secretory glands, which discharge a variety of odoriferous volatile chemicals that enable olfactory communication between conspecifics [13].

Apocrine glands are present in many ungulates, including sheep, horse, antelope, cow, and a few marsupials. They function as scent glands to produce pheromones [44]. The present study shows that the male and female interdigital glands have apocrine secretory lobules clustered in the dermal region. The myoepithelial cells observed in apocrine tubules produced evidence that the apocrine glands possess secretory cells. The myoepithelial cells surround the secretory acini, and may facilitate discharge of the secretory material. Yasui et al. [45] reported that the apocrine tubular glands found in the anal sacs secrete a substance containing complex glycolconjugates that facilitate the intraspecific communication as well as the signaling of sexual status. Japanese serow secretes large amounts of glycoconjugates (bound volatiles) through the apocrine portion of the infraorbital gland [46]. It is reported that the pheromone compounds derived mainly from the sebaceous glands, on discharge to the outside, can adhere to objects, and the secretion of apocrine glands promotes stronger adhesion capacity and high persistence in the scented site during the expression of territorial/scent marking [34,37,47]. The present result suggests that the apocrine gland in the interdigital gland of Vembur sheep contributes to the secretion of volatile substances that may be useful for chemical communication between Vembur sheep herds.

Pheromones are important molecules playing roles in animal communication, particularly breeding, territoriality, and conspecific recognition. In Vembur sheep, the interdigital gland is the major source of pheromones; it can be potentially involved in scent marking and convey various signals such as sexual attraction, aggression, etc. In the present study, 23 volatile compounds identified both on male and female interdigital gland posts were qualitatively different. One of the striking features of the identified volatile profiles in the present study is the presence of many alkane compounds as compared to other compounds such as carboxylic acid, alkene, and amine. Alkane compounds probably play a significant role in sexual attraction, as reported in several mammalian species [48,49].

The unique glandular volatiles, four from male (i.e., butanoic acid, 2-methylpropanoic acid, 1-heptanol and octadecanoic acid) and five from female (i.e., octane, 7-hexyl-tridecane, tetradecane, heptadecane and decanoic acid) identified in the present study are in the molecular weight range 88–284 Da and possess 4–19 carbon atoms. Pheromone compounds are usually in the molecular weight range 50–350 Da, possess 2 to 20 carbon atoms, and must be volatile to reach the receiver [50]. Octane and tetradecane are already reported in the interdigital glands of bontebok and blesbok antelopes [28]; 1-heptanol, decanoic acid, and octadecanoic acid are reported in the interdigital gland of bontebok- and blesbok antelopes [28] and the preorbital gland of blackbuck [38]; heptadecane in the preorbital glands of bontebok and blesbok antelopes [51]; and butanoic acid and 2-methylpropanoic acid in the interdigital gland of whitetail deer [29]; however, the function of these compounds has not been determined in these antelope species [28,29,38,51]. The roles of these compounds of male and female sheep require to be investigated further in order to find their contribution to the sheep’s behavior, which would provide additional insight into the olfactory communication (i.e., compounds that attract male/female) that could be used to improve the breeding management of Vembur sheep.

Further, the compounds, particularly, tetradecanol, tetradecanoic acid, and hexadecanol, having the highest peak concentration, are present in copious amounts in both males and females compared to other compounds. Tetradecanol and hexadecanol have been identified as the major compounds in the interdigital gland of pronghorn [30] but these three compounds have also been reported in the preorbital gland of blackbuck [38] and the interdigital gland of bontebok and blesbok antelopes [28]. It is interesting to note that the lone compound, 1-tetradecanol, has been reported as a growth inhibitor of the bacteria *Prionibacterium acnes* [52], *Bervibacterium ammoniagenes* [52] and *Streptococcus mutans* [53], and 1-hexadecanol is growth inhibitor of *P. acnes* [52]. The present study suggests that the sex-specific compounds would play a critical role in sexual attraction among Vembur sheep and, in addition, the high concentrations of these three compounds in both sexes may afford protection of the feet from microbial infection during poor hygienic conditions, especially in the rainy season and in mechanical injury, without the need of medication.

## 5. Conclusions

This is the first report on anatomy, morphology, histology, and volatile identification of the interdigital gland of the South Indian breed of Vembur sheep. The interdigital gland of male sheep is relatively large in size compared to the gland of female sheep, and it is made up of well-developed holocrine sebaceous and apocrine secretory lobules. Altogether, 23 volatile compounds were detected in glandular posts irrespective of sex, with 18 compounds found in the male sheep and 19 in the female sheep. Four compounds—butanoic acid, 2-methylpropanoic acid, 1-heptanol and octadecanoic acid- were specific to male glandular post, whereas octane, 7-hexyl-tridecane, tetradecane, heptadecane and decanoic acid showed to be specific for female gland secretion. In addition, the peak heights of tetradecanol, tetradecanoic acid, and hexadecanol, are high in both male and female sheep and they may be antibacterial compounds as in the interdigital gland of antelopes. Based on the morphometric and histomorphological observations, and the volatile characterization, we conclude that the Vembur sheep interdigital gland secretes chemical substances via sebaceous and apocrine secretory glands. These chemicals play a role in olfactory communication among animals as well as being antimicrobial agents that protect the feet against microbial infections. Further studies are required to determine if these three compounds possess anti-microbial properties in the context of common foot-borne diseases of sheep.

## Figures and Tables

**Figure 1 vetsci-09-00647-f001:**
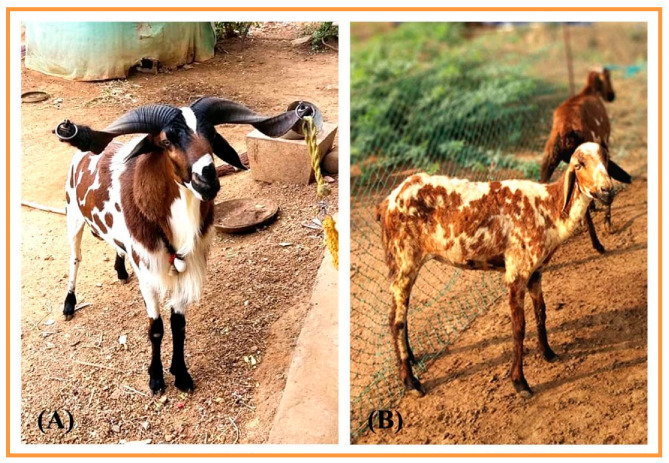
Vembur sheep: (**A**) male; (**B**) female.

**Figure 2 vetsci-09-00647-f002:**
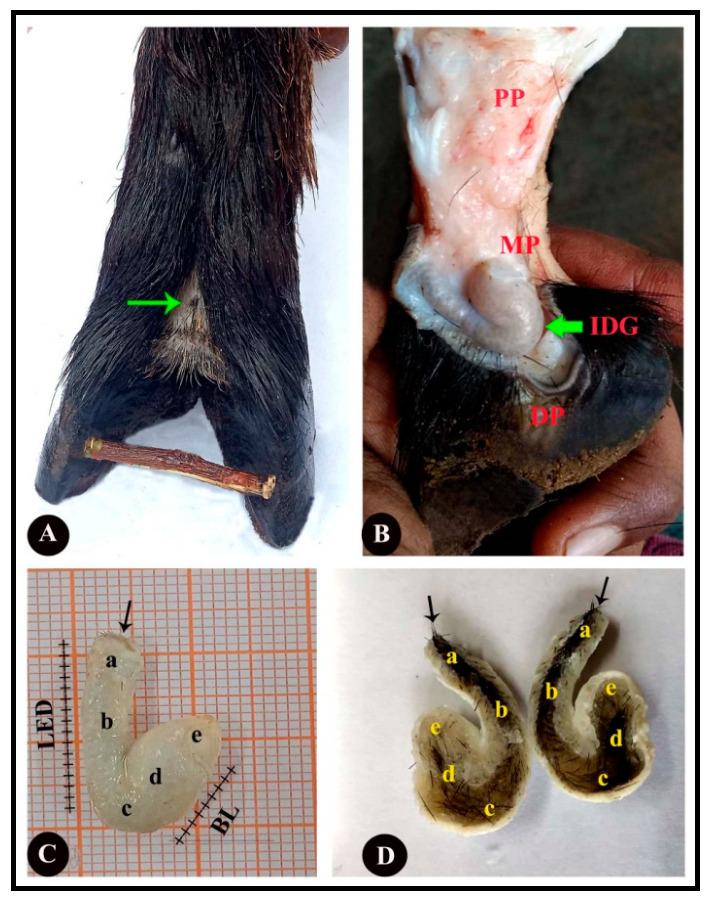
(**A**) Dorsal view of the interdigital region; the thin green arrow points to opening of the interdigital skin fold. (**B**) Medial view of the interdigital region; the thick green arrow points to the position of the sinus interdigitalis. IDG, interdigital gland. (**C**) The gland resembles a tobacco pipe in shape, with a wide body and long narrow neck. (**D**) Sagittal sectional view of the interdigital gland to show excretory duct filled with wooly fibers and its luminal content. a—orifice; b—excretory duct; c—flexure (distal part of the body); d—body of the sinus; e—fibrous attachment to the proximal part of the gland; BL—body length; LED—length of excretory duct; thin black arrow—external orifice; PP—proximal phalanx; MP—medial phalanx; DP—distal phalanx.

**Figure 3 vetsci-09-00647-f003:**
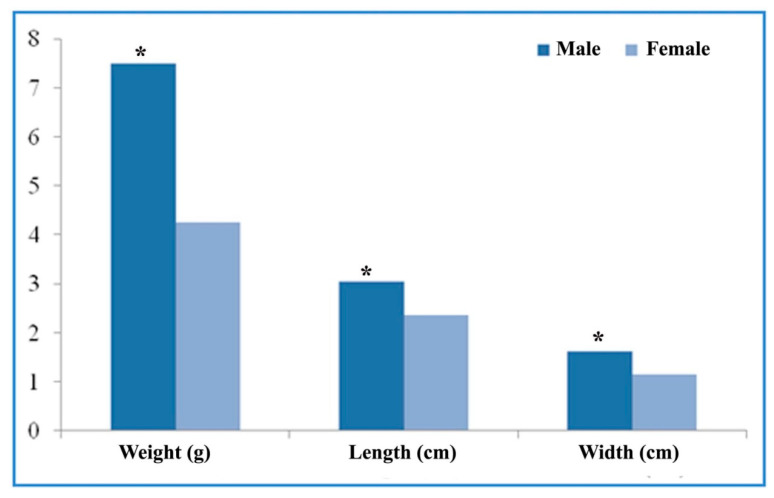
Mean values of the morphometric parameters (length, width, and weight) of male and female interdigital glands (CG). * *p* < 0.05, male gland compared to female gland.

**Figure 4 vetsci-09-00647-f004:**
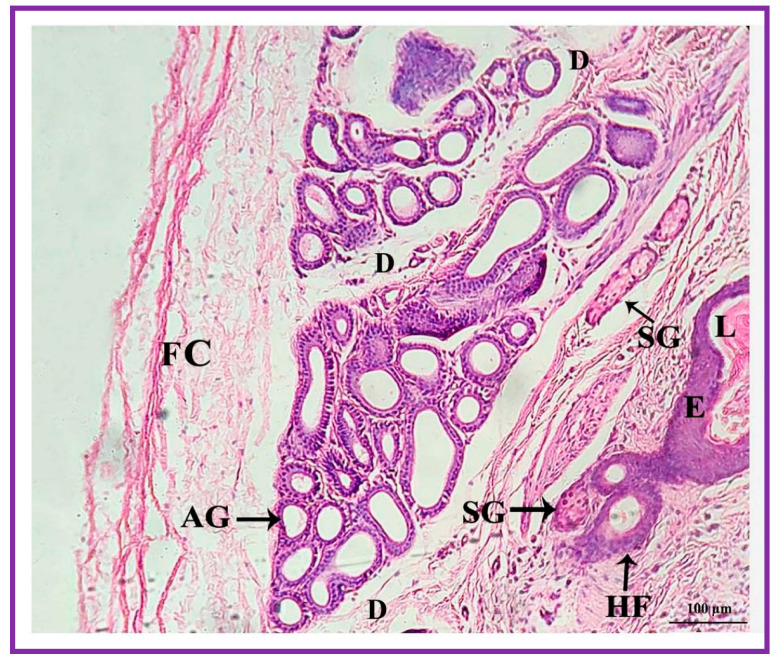
Photomicrograph of a section of the interdigital gland of Vembur sheep. The gland is ensheathed by fibrous capsule (FC); E: epidermis; D: dermis; HF: hair follicles; IL: interdigital lumen filled with waxy secretion; SG: sebaceous glandular portion; and AG: apocrine glandular portion; L: lumen.

**Figure 5 vetsci-09-00647-f005:**
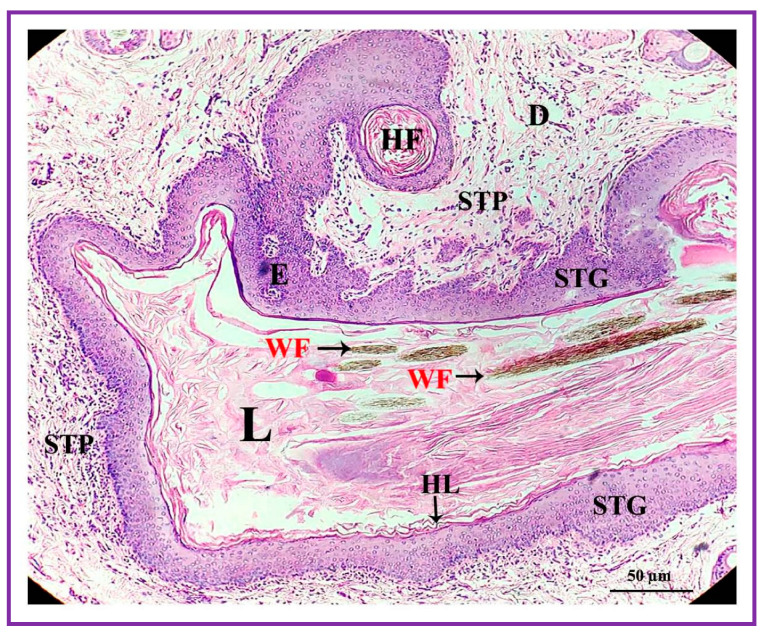
Epidermis (E) of interdigital gland with hyperkeratinized layer (HL), well developed stratum granular (STG) and stratum papillare (STP). The secretory material is present amidst and around the wooly fibers (WF) in the lumen (L). D: dermis; HF: hair follicle.

**Figure 6 vetsci-09-00647-f006:**
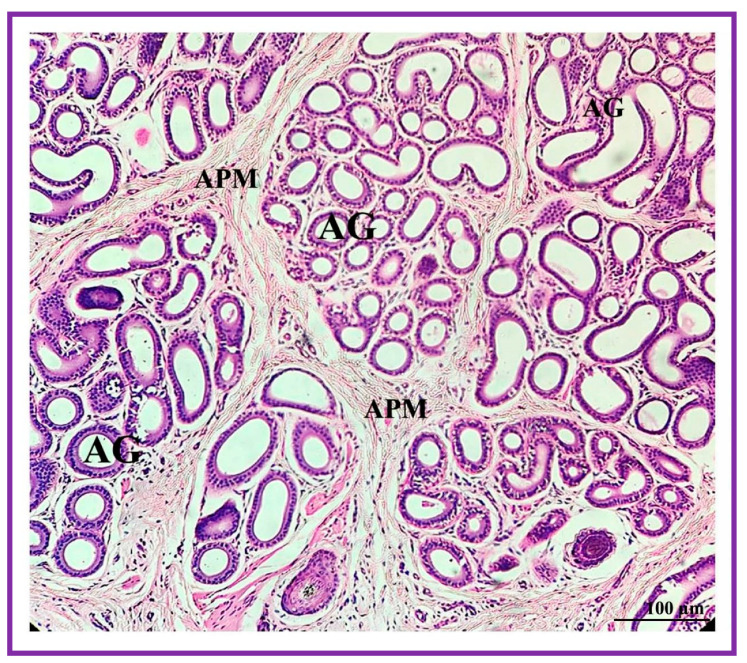
Photomicrograph of the interdigital sinus of male Vembur sheep showing clusters of apocrine secretory lobules (AG). Arrector pili muscles (APM) are found around the secretory glands (apocrine and sebaceous) and the hair follicles placed in the dermal region.

**Figure 7 vetsci-09-00647-f007:**
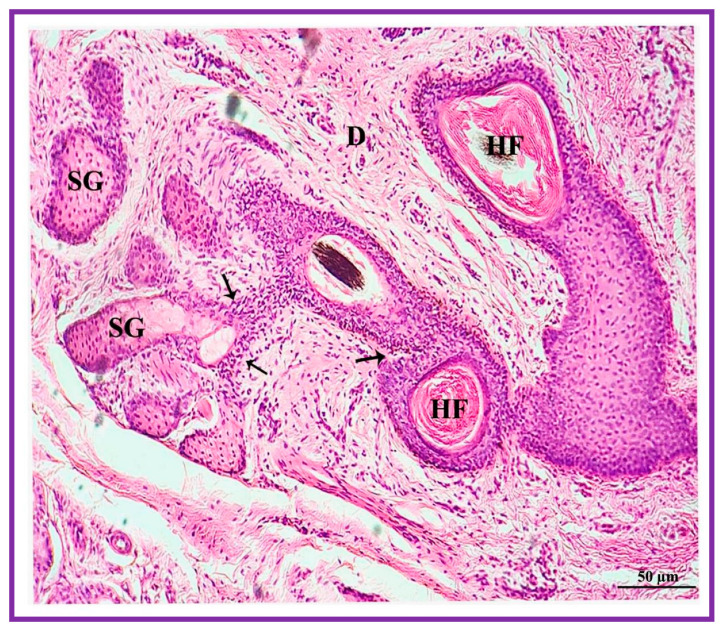
Photomicrograph of the interdigital sinus of Vembur sheep showing the branched excretory duct (thin arrow) of sebaceous glandular lobules (SG) connecting with hair follicles (HF). D: dermis.

**Figure 8 vetsci-09-00647-f008:**
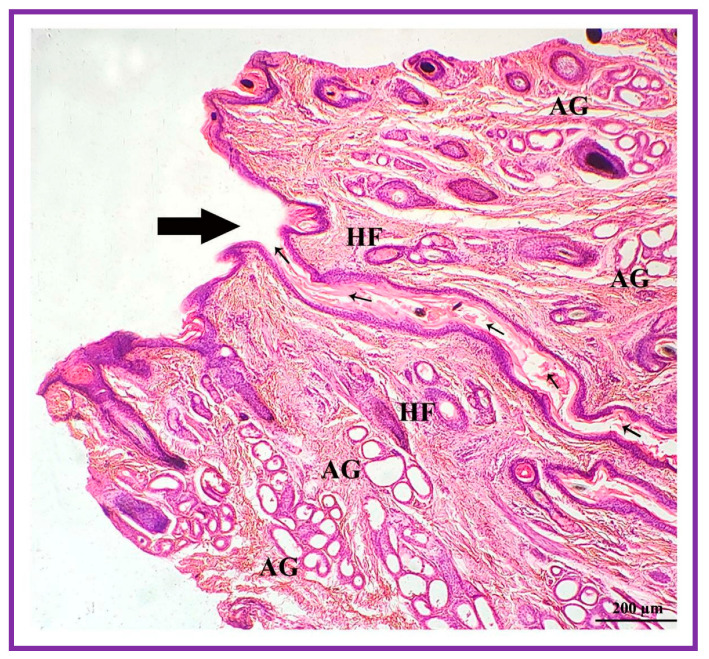
Glandular secretion (both apocrine and sebaceous) discharged into the narrow duct (thin arrow) which opens into the external orifice (thick arrow). AG: apocrine gland; HF: hair follicle.

**Figure 9 vetsci-09-00647-f009:**
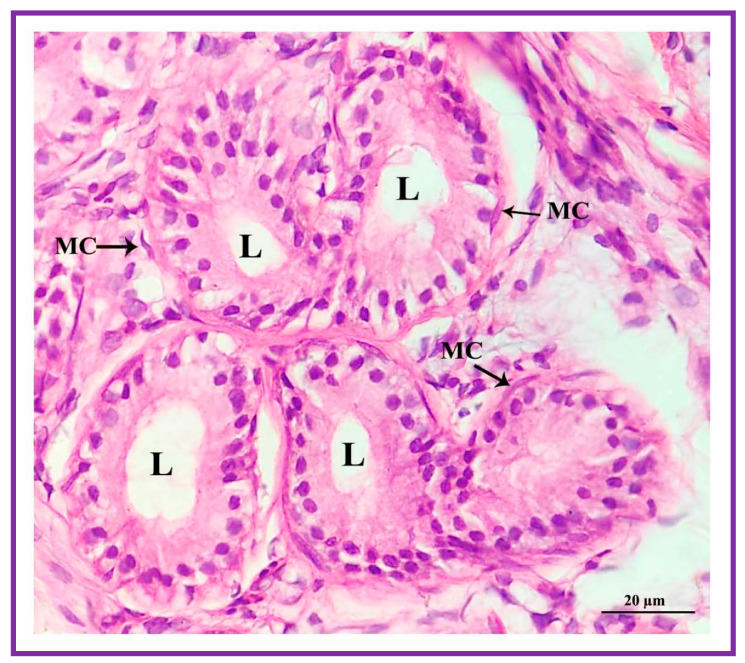
The apocrine gland is surrounded by collagenous or interlobular connective tissue of the hypodermis. The myoepithelial cells (MC) surround the apocrine tubules. L: lumen.

**Figure 10 vetsci-09-00647-f010:**
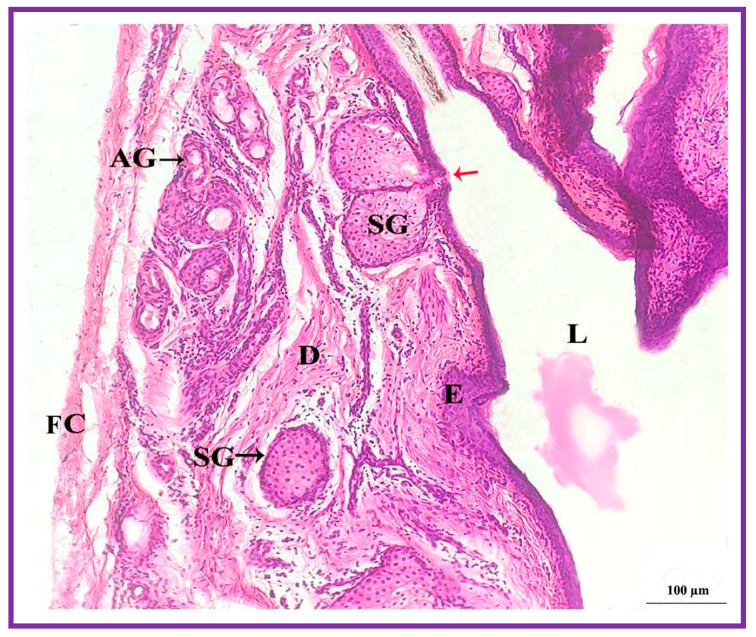
Photomicrograph of the interdigital sinus of male Vembur sheep showing the quite large holocrine sebaceous secretory lobules (SG) compared to apocrine gland (AG). Red arrow: secretion directly discharged into the lumen (L); FC: fibrous capsule; E: epidermis; D: Dermis.

**Figure 11 vetsci-09-00647-f011:**
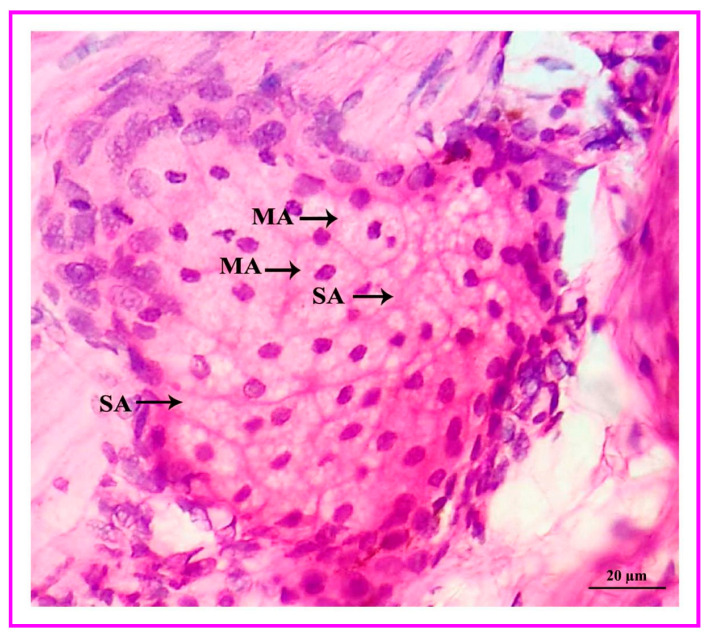
Simple alveolar holocrine sebaceous glandular lobules and the cytoplasm become honeycombed in appearance. MA: mucus acini; SA: serous acini.

**Figure 12 vetsci-09-00647-f012:**
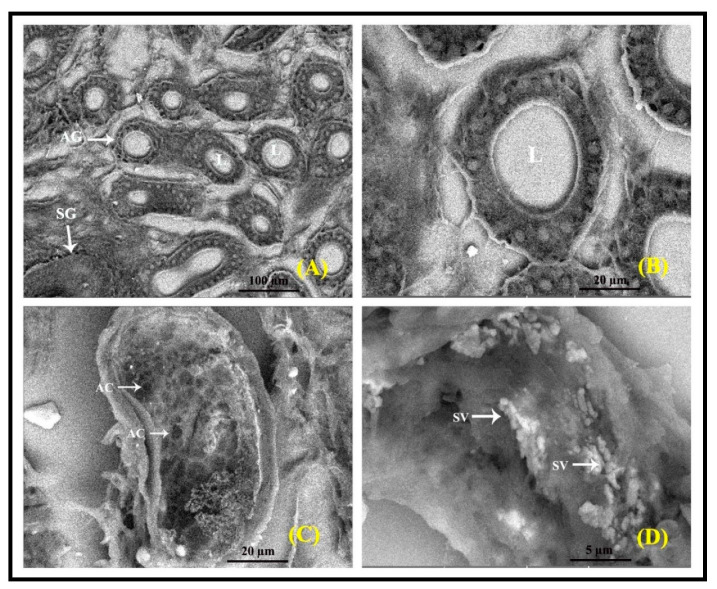
Scanning electron micrographs of the interdigital gland. The apocrine glandular lobules (AG) appear as a group of tortuous tubules found with sebaceous secretory lobules (**A**). The luminal surface shows a paved appearance (**B**); sebaceous glandular (SG) lobules with acinar cells (AC) (**C**); and secretory vesicles (SV) (arrows) with the secretory content present in the lumen (L) of the apocrine tubules (**D**).

**Figure 13 vetsci-09-00647-f013:**
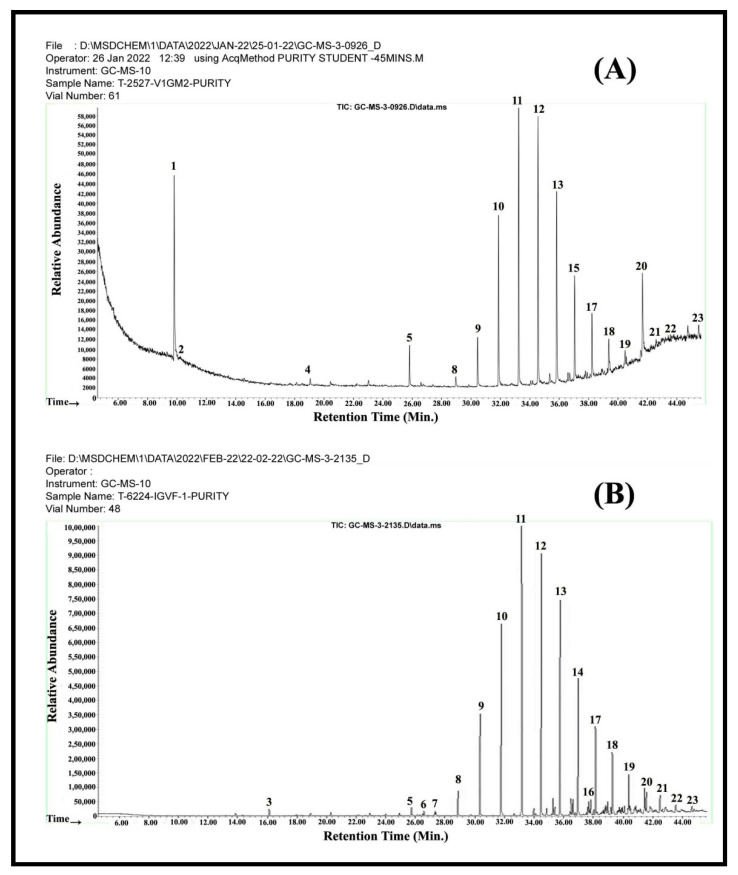
The gas chromatographic profiles of the volatile compounds of male (**A**) and female (**B**) interdigital gland secretion. For details, please refer Table 1.

**Table 1 vetsci-09-00647-t001:** The volatile compounds in the interdigital gland secretion of the male and female Vembur sheep.

Compound Name	Molecular Weight	Molecular Formula	Nature of Compound	Male		Female	
				Peak No.	% Peak Area	Peak No.	% Peak Area
Butanoic acid	88	C_4_H_8_O_2_	Carboxylic acid	1	12.06	-	-
2-Methylpropanoic acid	88	C_4_H_8_O_2_	Carboxylic acid	2	1.06	-	-
Octane	114	C_8_H_18_	Alkane	-	-	3	0.42
1-Heptanol	116	C_7_H_16_O	Alcohol	4	0.91	-	-
Decane	142	C_10_H_22_	Alkane	5	2.47	5	0.48
7-Hexyl- tridecane	172	C_10_H_20_O_2_	Alkane	-	-	6	0.27
Tetradecane	198	C_14_H_30_	Alkane	-	-	7	0.22
Dodecane	170	C_12_H_26_	Alkane	8	0.85	8	1.32
Pentadecane	212	C_15_H_32_	Alkane	9	3.45	9	5.00
2-Bromo-dodecane	249	C_12_H_25_Br	Alkane	10	9.64	10	11.02
Tetradecanol	214	C_14_H_30_O	Alcohol	11	14.16	11	15.91
Tetradecanoic acid	228	C_14_H_28_O_2_	Carboxylic acid	12	14.44	12	14.22
Hexadecanol	242	C_16_H_34_O	Alcohol	13	10.89	13	11.64
Heptadecane	240	C_17_H_36_	Alkane	-	-	14	7.51
Octadecanoic acid	284	C_18_H_36_O_2_	Carboxylic acid	15	6.12	-	-
Decanoic acid	268	C_19_H_40_	Carboxylic acid	-	-	16	0.91
Eicosane	282	C_20_H_42_	Alkane	17	4.11	17	5.13
1-Chloro-octadecane	288	C_18_H_37_Cl	Alkane	18	3.82	18	3.79
Heneicosane	296	C_20_H_44_	Alkane	19	2.04	19	2.65
Docosane	310	C_22_H_46_	Alkane	20	8.31	20	1.48
2-Cyclohexyl-eicosane	364	C_26_H_52_	Alkane	21	1.08	21	1.33
5-Butyl-docosane	366	C_26_H_54_	Alkane	22	0.94	22	0.83
Cholesterol	386	C_27_H_46_O	Steroid	23	1.67	23	0.21

## Data Availability

The data that support the findings of this study are embedded within the article.

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
