# Peer review of "Histomorphology and Chemical Constituents of Interdigital Gland of Vembur Sheep, Ovis aries"

_vetsci, 2022, doi:10.3390/vetsci9110647_

Round 1
Reviewer 1 Report
Article prepared by Thangavel Rajagopal et al. seems to be an interesting paper, presenting
both morphological and histological description of the interdigital glands of Vembur Sheep, in the context of possible sex dimorphism. An interesting elements is also description of the composition of the volatile compounds which could play a role in the process of chemical communication.
Authors with the use of proper techniques collected, evaluated and describe the structure of the gland documenting it with good quality, clearly described photos and figures. The paper is interesting and meets the criteria of a journal and the special issue. However some elements could be improved to make this manuscript more complete in my opinion.
I cannot find any information in the text about the physiology of reproduction of this breed. If they are not seasonal animals, that information must be included. If they are seasonal, information in which part of the year ( season ) glands were collected should be mentioned also.
The part of the season could heave influence on the composition of the secretion – it is not said that this compounds must be used in the context of reproductive behaviors, but we can not exclude it.
If the context of season was not included in this study it could be suggested that further study are also indicated in this context. Similarly the mentioned influence of hormones ( which were not examined directly in this particular study) could be examined in the further study.
Since we know that some hormones can influence the size of the glands and composition of the secretion ( what was mentioned in the text in the context of other glands present in sheep, as well as other glands in other species), the information about the status of the males also should be included. I guess, they were intact, but that must be clearly stated in the text if so, or the information about castrated males should be included. Levels of hormones could be seasonally dependent also.
I also suspect that all animals were adult but any information about that can’t be found in the text. In chapter Study animals authors describes the size of tail and ears but no information about the age of animals is present.
In description of the animals there is also lacking information how strong the sexual dimorphism is presented in this breed. We know that the ewes have polls whereas the males have horns, but what about the general size of the animals body- differences between males and females. If that difference is clearly marked, is it also reflected in the size of the limbs? If so, could it influence the general size of the glands ?
The font should be standardized throughout the manuscript. In some places (description of the substance) the font size seems to be different
Author Response
Review Report Form # 1
Comments and Suggestions for Authors
Article prepared by Thangavel Rajagopal et al. seems to be an interesting paper, presenting both morphological and histological description of the interdigital glands of Vembur Sheep, in the context of possible sex dimorphism. An interesting elements is also description of the composition of the volatile compounds which could play a role in the process of chemical communication.
Authors with the use of proper techniques collected, evaluated and describe the structure of the gland documenting it with good quality, clearly described photos and figures. The paper is interesting and meets the criteria of a journal and the special issue. However some elements could be improved to make this manuscript more complete in my opinion.
I cannot find any information in the text about the physiology of reproduction of this breed. If they are not seasonal animals, that information must be included. If they are seasonal, information in which part of the year (season) glands were collected should be mentioned also.
The part of the season could have influence on the composition of the secretion – it is not said that this compounds must be used in the context of reproductive behaviors, but we can not exclude it.
If the context of season was not included in this study it could be suggested that further study are also indicated in this context. Similarly the mentioned influence of hormones (which were not examined directly in this particular study) could be examined in the further study.
Author Response:
This has been included in Materials and Methods section of the revised manuscript. There are two breeding seasons in the Vembur breeding tract: most of the animals are in heat from March to May and a few are so from July to September. The samples were not taken seasonally but instead at random. The hormones, understandably, influence both the morphology and physiology of the interdigital glands. So, this subject will be covered in our future investigation.
Comment: Since we know that some hormones can influence the size of the glands and composition of the secretion (what was mentioned in the text in the context of other glands present in sheep, as well as other glands in other species), the information about the status of the males also should be included. I guess, they were intact, but that must be clearly stated in the text if so, or the information about castrated males should be included. Levels of hormones could be seasonally dependent also.
Author Response:
We concur with the Reviewer’s thoughts, but this study is overly focused on male-female differences in interdigital gland anatomy, morphology, and volatile compounds.
The even-toed ungulates (Artiodactyla) have many specialized skin glands, the secretions of which are used in semiochemical communication [5]. Variety of scent glands are present in ungulates as follows: sudoriferous glands are located between the antlers and eyes on the forehead; preorbital glands extend from each eye's medial canthus; nasal glands are found between the nostrils; preputial glands are found in the foreskin of the penis; interdigital glands are found between the toes; the perpetual glands are located within the penis foreskin; the metatarsal glands are located outside the hind legs; the tarsal glands are located within the hind legs; and the inguinal glands are located in the lower belly or groin [5].
This has been included in the introduction section of revised manuscript.
In our future research, we will pay attention to all concerns of the Reviewer.
Comment: I also suspect that all animals were adult but any information about that can’t be found in the text. In chapter Study animal’s authors describes the size of tail and ears but no information about the age of animals is present.
Author Response: The age of the animals are mentioned in the Chapter “ 2.2 Source of the interdigital gland”.
Comment: In description of the animals there is also lacking information how strong the sexual dimorphism is presented in this breed. We know that the ewes have polls whereas the males have horns, but what about the general size of the animals body- differences between males and females. If that difference is clearly marked, is it also reflected in the size of the limbs? If so, could it influence the general size of the glands?.
Author Response:
Male animals have far larger bodies than females, but this difference is not reflected in the size of their limbs or glands. During the study period, we noticed that most male animals had larger glands than female animals, but very seldom did we notice large glands in female animals. This could be due to hormones, age, seasons, etc.
Comment: The font should be standardized throughout the manuscript. In some places (description of the substance) the font size seems to be different.
Author Response:
This has been corrected throughout the manuscript.

Reviewer 2 Report
The Paper aimed at examining the sexual differences of interdigital glands in morphology, histology and chemistry of Vembur Sheep, Ovis aries. It is of significance to study the chemical signal producing organs and signal composition of this particular domestic sheep for understanding their basic biological properties and new farming improvements. I have some suggestions for improvement:
1. In INTRODUCTION, you should clearly state the significance of your work to the significance of farming, biology (e.g. sex recognition) and special significance to sheep, which have many odorant sources. Some chemical results from other related mamamals should been cited and stated here.
2. In METHODS, The absence of thorough chemical composition identification, that is, no comparison with authentic analogues, and the volatile compounds were temporarily identified should account for this defect.
3. In METHODS, In papreparing samples for GC-MS assay, it seems that a parallel control experiment should be there, due to a lot of saturated alkanes.
4. In RESULTS and DISCUSSION, In additon to sex-unique components, some quantitatively enhanced compounds in males or females might also be considered candidate pheromones. It seems that you did not perform such analysis and present such results, which should be explained in the discussion.
Author Response
Review Report Form # 2
Comments and Suggestions for Authors
The Paper aimed at examining the sexual differences of interdigital glands in morphology, histology and chemistry of Vembur Sheep, Ovis aries. It is of significance to study the chemical signal producing organs and signal composition of this particular domestic sheep for understanding their basic biological properties and new farming improvements. I have some suggestions for improvement:
Comment: 1. In INTRODUCTION, you should clearly state the significance of your work to the significance of farming, biology (e.g. sex recognition) and special significance to sheep, which have many odorant sources. Some chemical results from other related mamamals should been cited and stated here.
Author Response:
Information on chemicals from other / related mammals is covered under discussion (6th and 7th paragraphs). In the introduction, we have listed in the 4th paragraph the anatomical and chemical studies of several mammals.
Comment:
- In METHODS, The absence of thorough chemical composition identification, that is, no comparison with authentic analogues, and the volatile compounds were temporarily identified should account for this defect.
Author Response:
In this exploratory investigation, we used the standardized NICT 12 library that was built into the GC-MS system to identify the volatile chemicals of the interdigital gland post of male and female sheep. We did not compare the identified chemicals with authentic analogues.
We shall work on the Reviewer’s concerns in our future research.
Comment
- In METHODS, In papreparing samples for GC-MS assay, it seems that a parallel control experiment should be there, due to a lot of saturated alkanes.
Author Response:
We did not conduct a control experiment in parallel with the GC-MS assay. Our analysis was limited to GC-MS of samples from adult males and females. We'll take the Reviewer’s suggestions into account in our upcoming research.
Comment
- In RESULTS and DISCUSSION, In additon to sex-unique components, some quantitatively enhanced compounds in males or females might also be considered candidate pheromones. It seems that you did not perform such analysis and present such results, which should be explained in the discussion.
Author Response:
This has been incorporated in the revised manuscript (6th paragraph).
GENERAL: Thanks to the observation of Reviewer 2, we have further critically improved the English language, so the article now reads much better.
